



# Measurement Report: Tropospheric and Stratospheric Ozone Profiles during the 2019 TROpomi vaLIdation eXperiment (TROLIX-19)

John T. Sullivan[1], Arnoud Apituley[2], Nora Mettig[3], Karin Kreher[4], K. Emma Knowland[1,5], Marc Allaart[2], Ankie Piters[2], Michel Van Roozendael[6], Pepijn Veefkind[2], Jerry R. Ziemke[1,5], Natalya Kramarova[1], Mark Weber[3] Alexei Rozanov[3], Laurence Twigg[1,7], Grant Sumnicht[1,7], and Thomas J. McGee[1*]

[1] NASA Goddard Space Flight Center, Greenbelt, MD 20771
[2] Royal Netherlands Meteorological Institute (KNMI), De Bilt, Netherlands
[3] Institute of Environmental Physics, University of Bremen, Bremen, Germany
[4] BK Scientific GmbH, Mainz, Germany
[5] Morgan State University/GESTAR-II, Baltimore, MD 21251
[6] Belgian Institute for Space Aeronomie (BIRA), Ukkel, Belgium
[7] Science Systems and Applications Inc., Lanham, MD, 20706
*Now Emeritus

*Correspondence to*: John Sullivan (john.t.sullivan@nasa.gov)

**Abstract** A TROPOspheric Monitoring Instrument (TROPOMI) validation campaign was held in the Netherlands based at the CESAR (Cabauw Experimental Site for Atmospheric Research) Observatory during September 2019. The TROpomi vaLIdation eXperiment (TROLIX-19) consisted of active and passive remote sensing platforms in conjunction with several balloon-borne and surface chemical (e.g. ozone and nitrogen dioxide) measurements. The goal of this joint NASA-KNMI geophysical validation campaign was to make intensive observations in the TROPOMI domain in order to be able to establish the quality of the L2 satellite data products under realistic conditions, such as non-idealized conditions with varying cloud cover and a range of atmospheric conditions at a rural site. The research presented here focuses on using ozone lidars from NASA's Goddard Space Flight Center to better evaluate the characterization of ozone throughout TROLIX-19. Results of comparisons to the lidar systems with balloon, space-borne, and ground-based passive measurements are shown. In addition, results are





compared to a global coupled chemistry meteorology model to illustrate the vertical variability and
columnar amounts of both tropospheric and stratospheric ozone during the campaign period.

## 1 Introduction

In September 2019, a joint Royal Netherlands Meteorological Institute (KNMI) and the U.S. National
Aeronautics and Space Administration (NASA) field campaign was performed in the Netherlands,
based at the Cabauw Experimental Site for Atmospheric Research (CESAR, 51.97° N, 4.93° E), to
provide the scientific community with additional information to further understand and evaluate the
Copernicus Sentinel-5 Precursor mission (S-5P) TROPOspheric Monitoring Instrument (TROPOMI)
instrument (https://sentinels.copernicus.eu/web/sentinel/missions/sentinel-5p). The main objective of
the Copernicus Sentinel-5P mission is to perform atmospheric measurements with high spatio-temporal
resolution, to be used for scientific studies and monitoring of air quality and chemical transport
(https://www.esa.int/Applications/Observing_the_Earth/Copernicus/Sentinel-5P).
To properly support satellite evaluation, the 2019 TROpomi vaLIdation eXperiment (TROLIX-19)
campaign was designed to bring together many active and passive remote sensing platforms in
conjunction with several balloon-borne, airborne and surface measurements. Specifically, the
observations were established to provide geophysical verification in order to establish the quality of
TROPOMI Level 2 (L2) main data products under realistic non-idealized conditions with varying cloud
cover and a wide range of atmospheric conditions. Cabauw, using its comprehensive in-situ and remote
sensing observation program in and around the 213 m meteorological tower (https://ruisdael-
observatory.nl/trolix19-tropomi-validation-experiment-2019/) was the main site of the campaign with



focus on vertical profiling using lidar instruments for aerosols, clouds, water vapor, tropospheric and
stratospheric ozone, as well as balloon-borne sensors for nitrogen dioxide ($NO_2$) and ozone (Figure 1).
Although this work focuses primarily on the ozone lidar profiling during the study, the larger campaign
overview, background, and motivation can be found in Apituley *et al.* (2019; 2020) or Kreher *et al.*
(2020a).
One main goal of this work is also to understand ozone profile retrievals as they relate to upcoming
satellite endeavors. As NASA prepares to launch its first geostationary air quality satellite
"Tropospheric Emissions: Monitoring of POllution" (TEMPO), this work also specifically establishes a
paradigm of evaluation for TEMPO-derived products such as tropospheric ozone columns and a 0-2km
tropospheric ozone product. Due to the finer spatial footprint, increased temporal frequency and vertical
extent of TEMPO's tropospheric ozone retrievals, ozone lidars are an ideal platform to perform future
evaluations of the products, which builds from recent work done in Johnson *et al.,* 2018. Specifically,
this work will investigate the results from the combination of having a co-located NASA tropospheric
(Sullivan et al., 2014) and stratospheric ozone lidars (McGee et al., 1991) in order to obtain an entire
vertical profile of ozone from ~0.2km to 50km.
For the first time, this transportable combination of lidars is able to explicitly derive diurnally varying
tropospheric and total ozone columns, which are compared directly to measurements obtained by
ground-based passive sensors, current satellite instrumentation and chemical transport models. In
Section 2 we present all available data and methods used in this work, across the various platforms
during the TROLIX-19 study. Section 3 focuses on comparisons of the tropospheric ozone retrievals of
the vertical profiles of ozone within the troposphere and columnar reductions of 0-10 km and 0-2 km.



Comparisons of lidar data with available complete ozone profiles (Sec 4) and columnar amounts (Sec 5)
from several platforms and chemical transport models are also presented to further understand the
quality of satellite derived ozone profiles during the TROLIX-19 period.

2  Data and Methods
Descriptions of the various observational and model data sets and used in this study are below,
including a summary table (Table 1).
**2.1 NASA Ozone Differential Absorption Lidars (DIAL)**
NASA deployed and operated two ozone lidars during TROLIX-19 at the Cabauw site near the CESAR
tower to observe temporal and vertical gradients in tropospheric and stratospheric ozone.  This was the
first dual-deployment of these lidars, in which the tropospheric ozone lidar measured between the near
surface (about 0.2 km) to a height of about 18 km and the stratospheric lidar during night-time from 15
km upwards to nearly 50 km, providing complete hybrid ozone profiles for the campaign period.
The NASA GSFC Mobile Stratospheric Ozone Lidar Trailer Experiment (STROZ-LITE) has been a
participant in the Network for the Detection of Atmospheric Composition Change (NDACC) since its
inception and is housed in a 12.5m container allowing for transport around the world (McGee et al.,
1991). The lidar instrument transmits two wavelengths, 308 nm from a XeCl excimer laser, and 355 nm
from a ND:YAG laser to derive ozone number density profiles, which have historically served as an
intercomparison data set for other NDACC ozone lidars (recent intercomparison can be found at Wing
*et al.,* 2020; 2021).



The NASA GSFC TROPOZ has been developed in a transportable 13.5m trailer to take routine
measurements of tropospheric ozone near the Baltimore–Washington, D.C. area as well as various
campaign locations (Sullivan *et al.,* 2014; 2015,2019, Leblanc *et al.*, 2018). This instrument, which
utilizes a ND:YAG laser and Raman cell,  has been developed as part of the ground-based Tropospheric
Ozone Lidar NETwork (TOLNet, https://www-air.larc.nasa.gov/missions/TOLNet/), which currently
consists stations across the North America (http://www-air.larc.nasa.gov/missions/TOLNet/). The
primary purposes of the instruments within TOLNet are to provide regular, high-fidelity profile
measurements of ozone within the troposphere for satellite and model evaluation. This lidar also
operates routinely for the Network for the Detection of Atmospheric Composition Change (NDACC).
Both lidars collect backscattered radiation with a large primary telescope and a 10cm telescope for near
field channels. Spectral separation is accomplished using dichroic beam-splitters and interference filters.
For the stratospheric system, five return wavelengths are recorded: the two transmitted wavelengths,
and the nitrogen Raman scattered radiation from each of the transmitted beams 332 nm and 382 nm, and
the 408 nm water vapor channel. In this arrangement for TROLIX-19, the tropospheric system pumped
the Raman cell with the fourth harmonic (266 nm), which resulted in conversion to 289 nm and 299 nm
using a single hydrogen/deuterium Raman cell. All of the signals are further split to improve the
dynamic range of the respective lidar optical detection chains and are then amplified, discriminated and
recorded using photon counting techniques.
During TROLIX-19, the STROZ-LITE was operated on cloud free nights, with measurements lasting
between 2-4 hours to obtain enough signal to properly retrieve the entire stratospheric ozone profile.
The TROPOZ was operated during daytime and night time to provide tropospheric ozone profiles. For





instances of TROPOMI overpasses, campaign ozonesondes, or coincident stratospheric ozone lidar
measurements, the TROPOZ reported data is averaged for 30 minutes, centered around the satellite
overpass or launch time. This temporal period of averaging has been optimized in several cases to avoid
cloud contamination. For all other times during the TROPOZ operation, the data has been averaged to
10 minutes, which is suitable under most clear sky conditions to retrieve ozone information within the
entire troposphere.
**2.2  Ground Based Passive Sensors and Ozonesondes**
**2.2.1 Pandora Spectrometer Instrument**
A Pandora spectrometer instrument (#118) has been used to measure columnar amounts of trace gases
in the atmosphere at 3–5-minute resolution at the Cabauw site since 2016 and previously used for the
second Cabauw Intercomparison of Nitrogen Dioxide (CINDI-2) campaign (Kreher et al., 2020b).
Using the theoretical solar spectrum as a reference, Pandora determines trace gas amounts using
differential optical absorption spectroscopy (DOAS). This attributes in principal these differences in
spectra measured by Pandora to the presence of trace gases within the atmosphere (*i.e.* the difference
between the theoretical solar spectrum and measured spectrum is caused by absorption of trace gas
species). For this study, L2 direct sun columnar values of ozone are used, although retrievals of nitrogen
dioxide are also operationally acquired. Data used passed the strictest QC/QA estimate (Flags = 10) and
was obtained from the Pandonia Global Network (http://data.pandonia-global-network.org/).



### 2.2.2 Brewer MKIII Spectrophotometer

A Brewer MKIII spectrometer instrument (#189) has been used to measure daily columnar amounts of ozone in the atmosphere at the KNMI/De Bilt (30km NE of Cabauw) site since 2007. The Brewer is specifically designed to provide high accuracy measurement of spectrally resolved UV for satellite evaluation, climatology monitoring and public health to international standards. Similar to Pandora spectrometers, these measurements of total column of trace gases are compared to the measured UV spectrum with the known solar output, and modeling the scattering properties of the atmosphere and have been historically used to evaluate columnar satellite products (McPeters *et al.,* 2007; Wenig *et al.,* 2008; Garane et al., 2019). The Brewer is the standard instrument used in the World Meteorological Organization ozone monitoring network and for NDACC. This data was obtained at the NDACC website (https://www-air.larc.nasa.gov/missions/ndacc/data.html).

### 2.2.3 Ozonesondes

Ozonesondes have been used to measure vertical profiles of ozone in the atmosphere at the KNMI/De Bilt (30km NE of Cabauw) site since November 1992, and measurements are made weekly, historically at 12 UTC on Thursdays. Description of the Electro Chemical Cell (ECC) details and metadata are summarized in Malderen et al., 2016, which also describes the importance of understanding and reporting changes in ozonesonde operation procedures.  During the campagin, in situ measurements of ozone were made using a balloon-borne payload consisting of an ECC ozonesonde (Science Pump Corporation) coupled with a radiosonde and have been used to evaluate TROPOMI tropospheric ozone products in the tropics (Hubert et al., 2021). The ECC technique is widely used for the high vertical



resolution measurements of O₃. The ECC consists of two chambers with platinum electrodes immersed
in potassium iodide (KI) solutions at different concentrations. The accuracy in the O₃ concentration
measured by an ECC ozonesonde is ± 5%–10% up to an altitude of 30 km (Smit et al., 2007). This data
was obtained at the NDACC website (https://www-air.larc.nasa.gov/missions/ndacc/data.html).
**2.3 Satellite Observations and Products**
Satellite data used in this work was selected based on the closest retrieval (*i.e.* column, profile) to the
CESAR station within +/-2.5 degrees latitude and +/-10 degrees in longitude.
**2.3.1 Ozone Mapping and Profiling Suite (OMPS) and MERRA-2 products**
Daily total column ozone overpasses over Cabauw station from the Ozone Mapping and Profiling Suite
(OMPS) Nadir-Mapper (NM) instrument on the Suomi National Polar-orbiting Partnership (S-NPP)
platform areused in this study. The vertical distribution of ozone in the stratosphere and lower
mesosphere is obtained from the OMPS Limb-Profiler (LP) sensor on the Suomi-NPP satellite merging
the UV (29.5-52.5 km) and VIS (12.5-35.5 km) bands to provide a full profile from 12.5km to 52.5km
(Kramarova *et al*., 2018). Variations of this merged OMPS-LP retrieval were considered, however the
work shown in Arosio *et al.,* 2018, indicates the same overall conclusions would be reached. Further
work beyond this manuscript may involve comparing this TROLIX-19 measurement data set to specific
experimentally performed satellite retrievals.
The Modern-Era Retrospective analysis for Research and Applications, Version 2 (MERRA-2) provides
data beginning in 1980 and since August 2004 assimilates NASA's satellite ozone profile observations



from Aura Microwave Limb Sounder (MLS) (Livesey et al, 2008) to more comprehensively
characterize stratospheric ozone abundance. A residual tropospheric ozone product (Ziemke *et al.,*
*2019*) is derived using the OMPS NM total column ozone minus the co-located MERRA-2 stratospheric
column ozone. Tropopause pressure is derived from MERRA-2 potential vorticity (2.5 PVU) and
potential temperature (380 K).
**2.3.3 MLS**
NASA's Aura Microwave Limb Sounder (MLS) uses microwave emission to measure stratospheric
and upper tropospheric constituents, such as ozone. Ozone data (v5) used in this study is binned on
various vertical grids and are converted from volume mixing ratio to number density using the
coincident MERRA-2 atmosphere state parameters. Both daytime and nighttime data are used in this
study and the corresponding closest profile is utilized for comparison.
**2.3.4 TROPOMI**
In October 2017, the Sentinel-5 Precursor (S5P) mission was launched, carrying the TROPOspheric
Monitoring Instrument (TROPOMI), which is a nadir-viewing 108° Field-of-View push-broom grating
hyperspectral spectrometer. Starting in August 2019, Sentinel-5P TROPOMI along-track high spatial
resolution (approximately 5.5 km at nadir) has been implemented and total ozone columns values used
in this work are subsetted from the NASA GES DISC
(https://tropomi.gesdisc.eosdis.nasa.gov/data/S5P_TROPOMI_Level2/S5P_L2__O3_TOT_HiR.1/) to
provide the Offline 1-Orbit L2 (S5P_L2__O3_TOT_HiR), which is based on the Direct-fitting



algorithm (S5P_TO3_GODFIT), comprising a non-linear least squares inversion by comparing the
simulated and measured backscattered radiances.
Tropospheric Ozone vertical profiles were retrieved using the TOPAS (Tikhonov regularized Ozone
Profile retrievAl with SCIATRAN) algorithm and were applied to the TROPOMI L1B spectral data
version 2, using spectral data between 270 and 329 nm for the retrieval (Mettig *et al.,* 2021). This data
set will cover the TROLIX-19 period from 09 September until 28 September; however, it is available
outside of this work for specific weeks between June 2018 and October 2019. Since the ozone profiles
are very sensitive to absolute calibration at short wavelengths, a re-calibration of the measured
radiances is required using comparisons with simulated radiances with ozone limb profiles from
collocated satellites used as input. The a priori profiles for ozone are taken from the ozone climatology
of Lamsal et al. (2004) and the calibration correction spectrum is determined using the radiances
modelled with ozone information from collocated MLS/Aura measurements as described in depth
throughout Mettig *et al.,* 2021.

**2.4 Coupled Chemistry and Meteorology Model**
The GEOS Composition Forecasting (GEOS-CF,
https://gmao.gsfc.nasa.gov/weather_prediction/GEOS-CF/, Keller *et al.,* 2021, Knowland et al., 2021)
system was chosen to serve as a comparison simulation for this effort, based on its altitude coverage (up
to 80 km) and implications for future geostationary satellite use. The system produces global, three-
dimensional distributions of atmospheric composition with a spatial resolution of 25km. Using
meteorological analyses from other GEOS systems, the GEOS-CF products include a running



atmospheric replay to provide near-time estimates of surface pollutant distributions and the composition
of the troposphere and stratosphere. Individual case study evaluations using ozone lidar of the GEOS-
CF meteorological replay have recently been performed in Dacic et al. (2020), Gronoff et al. (2021) and
Johnson et al. (2021). These results will also be used to better evaluate the GEOS-CF as the source of a
priori ozone profiles for use in the TEMPO tropospheric ozone retrievals. Model output for this work is
used from the closest GEOS-CF model grid cell to the CESAR observatory.

## 3  Tropospheric Ozone Comparisons

### 3.1 Vertical Profiles

Example tropospheric ozone profile observations are presented in **Figure 2** for 7 individual observation
periods during the TROLIX-19 campaign. Each of the panels show the cloud screened TROPOZ lidar
retrievals (top panels) and the corresponding GEOS-CF model output (bottom panels). Pink dots are
overlaid to indicate the simulated tropopause altitude based on a blended estimate (TROPPB) which
meets criteria of the lowest altitude bin corresponding with either a pressure level above the thermal
tropopause (380K) or dynamical (3 PVU) tropopause.
In general, the observations and simulations agree quite well in characterizing the broad features that
impacted the CESAR site during the TROLIX-19 campaign. However, in each panel there are ozone
laminae within the lower troposphere that are not replicated in the model simulation, most notably
during the September 20-21 period (Figure 2d-e). The observations indicate increased  ozone levels as
compared to the model during this period, centered around the 3-5km and 8-10km region of the



atmosphere (this is explored in more detail below). However, the model does simulate well the lowered
tropopause height and abundance of lower stratospheric ozone observed in the 2 October observations
(see Fig. 2g), which is an indication of the model well representing the dynamical variability that affects
the lowering of the tropopause height.
To bring in additional platforms and to better understand these differences throughout the campaign at
discrete altitudes, **Figure 3** shows the ozone number density values for the TROPOZ lidar, GEOS-CF
model, TROPOMI and ECC ozonesondes at the 4 km vertical level for the entire TROLIX-19 campaign
period. Within the 4km layer, the platforms are all characterizing the general ozone features throughout
the campaign at an altitude that frequently is associated with aged transported layering. There is a
noticeable difference between the observations and model during the previously described 20-21
September period. On 21 September at 12 UT, the lidar and ECC sonde quantify an elevated layer (1.2-
1.3 x $10^{21}$ molecules m$^{-3}$)  into the region that is not simulated by model (0.75-0.9 x$10^{21}$ molecules m$^{-3}$),
resulting in an approximately 30% difference in ozone abundance within layer. Since the model
correctly simulated many other ozone features during this time period within the upper tropospheric
region, this is likely aged transport into the domain that was not available during model initialization.
Back-trajectories were performed to better identify the source of this air mass, however nothing
conclusive can be reported. The layer is not associated with any increase in lidar attenuated backscatter
within the associated altitude, suggesting it was not urban in origin and therefore more likely aged
stratospheric air mixing down to the lower free troposphere. Outside of this Sep 21 period, there is
generally good agreement between the observations and model, indicating the combination of



observations and modeling are able to represent the rural conditions and ozone perturbations at the
CESAR site.

## 3.2 Columnar Data Reduction

There continues to be a need within the atmospheric and satellite community to understand the
variability of ozone as it pertains to both the tropospheric column (*i.e.* the Earth's surface to the
tropopause height) and the 0-2km tropospheric column (*i.e.* the Earth's surface to the 2 km height). The
0-2 km region is of particular interest as it is projected to be delivered hourly from the North American
geo-stationary satellite: Tropospheric Emissions: Monitoring of Pollution (TEMPO). Due to the
increased temporal frequency and vertical extent of TEMPO's tropospheric ozone retrievals, ozone
lidars, such as those from TOLNet (https://www-air.larc.nasa.gov/missions/TOLNet/) used in this work,
are an ideal platform to perform future evaluations of the products, which build from Johnson *et al.,*

58    2018.

Full tropospheric columns (**Figure 4, top panel**) are consistently calculated from each platform using
the blended tropopause height (TROPPB) produced by the GEOS-CF and described above (*c.f.* pink
dots in **Figure 2**) and are then converted to Dobson Units (DU). The tropospheric columns are
calculated explicitly by integrating the ozone number density from the lowest data bin of usable data to
the TROPPB produced in the nearest model temporal output. The exception to this is the
OMPS/MERRA-2 tropospheric column using the residual method described above (subtracting the
MERRA-2 stratospheric column from the OMPS-NM total ozone column). For the 0-2km tropospheric





column (**Figure 4, bottom panel**), there were no major surface layer pollution events at the CESAR
observatory during TROLIX-19.
For the full tropospheric column (Figure 4, top panel), the campaign variability ranges from
approximatively 20-55 DU. The model, lidar, and ECC sonde observations agree quite well throughout
the 12 Sep to 23 Sep time frame when looking at day-to-day variability. However, when assessing the
variability on a single day for 21 Sep, full tropospheric columns reduced from the lidar observations are
some of the largest observed during this TROLIX-19 period (reaching nearly 50 DU), while the model
mainly ranges between 35-37 DU or a difference of upwards of 40%. Therefore, this increase identified
and discussed in Fig 3, not only results an altitude specific difference, but ultimately results in a large
overall impact to the full tropospheric column content. This suggests ground-based profiling
observations are still critically needed to confirm large deviations from a priori and climatology in order
to evaluate the atmospheric chemistry models.
There exists both diurnal and day-to-day variability of the 0-2 km ozone, ranging from 4-10 DU
(Figure 4, top panel). In the 0-2 km ozone reduction, the lidar and model are critically needed to
understand ozone variability on a continuous scale. For instance, on 15 Sep the 0-2 km ozone column
was near 9 DU at 03 UT and finished near 5.5 DU at 16 UT, resulting in a -60% change in DU within a
single day. Furthermore, the gradient of the 21 Sep ozone column change was similar in scale to the
entire campaign variability, indicating that there is a significant amount of information gained in the
understanding of the variability in ozone from continuous measurements. Although a daily snapshot of
OMPS-MERRA-2 residuals and TROPOMI ozone profile observations are critical for their vast spatial





coverage, ground-based observations such as ozone lidar and ECC sondes are critically needed to
quantify measurement gaps.
In summary, we find that the ozone columns evaluated in this study generally reproduced the structure
of the TROLIX-19 ozone lidar observations for N=835 coincidences. For the full tropospheric column,
the lidar calculated median was 30.9 ± 4.7 DU, compared to 33.4 ± 3.9 DU for the GEOS-CF. This
indicates a difference of 2.5 DU or 7.9 %, which is well within the lidar uncertainty of around 10 %
throughout the tropospheric column, and as we described above is likely driven by select days rather
than an overall bias between the measurements.. For the 0-2 km tropospheric column, the lidar
calculated median was 5.8 DU ± 0.9 DU, compared to 6.9 DU ± 0.7 DU for the corresponding GEOS-
CF measurements. This indicates a difference between observations and model of 1.1 DU or 18.9 %,
which is higher than the lidar uncertainty of around 5-10 % throughout the column. For the TROLIX-19
campaign, a 0-2 km tropospheric column accounts for approximately 20% of the tropospheric column
as detailed in Figure 4 (top panel), indicating measurements above the surface are critically needed at
understanding ozone variability at rural sites such as Cabauw, NL, where free tropospheric ozone
features dominate the column.
**4 Full Profile Ozone Comparisons**
**4.1 Hybrid Tropospheric/Stratospheric Ozone Comparisons**
To better understand differences in ozone retrievals from multiple platforms, it is important to assess the
entire vertical distribution of ozone. To characterize the vertical distribution throughout the entire





troposphere and stratosphere, hybrid ozone profiles were created from longer (integrations of 60-120
minute vs 10 minutes in Sec 3) temporal retrievals from the co-located TROPOZ and STROZ lidars,
which were then interpolated to the GEOS-CF model vertical grid levels. **Figure 5** compares these
results to the GEOS-CF, OMPS-LP, TROPOMI, MLS and the ECC ozonesonde profiles for 12 Sep, 17
Sep, 19 Sep, and 21 Sep 2019. These days were selected as days within the campaign that had an ECC
launch from De Bilt, NL (30 km from Cabauw).
For each observation period in **Figure 5**, all platforms manage to characterize a similar shape and extent
of the ozone maxima between 2.5-4.5 molecules m$^{-3}$ throughout the vertical layer between 20-25 km. In
each case, there are differences between the platforms in characterizing the vertical variability and
extent of the ozone maxima, which will be quantified in the following section. One notable feature that
emphasizes the cross-platform ability to illustrate ozone variability in the stratosphere is from the 19
and 21 Sep profiles. A dual ozone maximum is observed quite remarkably by the merged lidar, ECC,
MLS, OMPS-LP and simulated by the GEOS-CF centered around 20 km and then again at 25km. The
wind observations from the ozonesonde payload (not shown) indicate a wind shear within the two ozone
layers, suggesting this feature was dynamically driven. The TROPOMI retrieval is not able to retrieve
this vertical features due to its coarser vertical resolution and appears to average through the layers.
**4.2 Difference Profiles**
To quantitatively compare the ozone retrievals and simulations, **Figure 6** displays the ozone
values for the TROLIX-19 time period from the hybrid lidar dataset (**Figure 6a**), GEOS-CF (**Figure**
**6b**), OMPS-LP (**Figure 6c**), MLS (**Figure 6d**) and TROPOMI (**Figure 6e**). This double ozone maxima,



starting after 20 September serves as a geophysical marker to visually compare the ozone products. The
lidar, model, and OMPS-LP all capture this feature, but with varying ozone abundances and altitudes.
From **Figure 6d**, it appears as if TROPOMI retrievals are not able to resolve this feature. The percent
differences, as compared to the lidar observations, are displayed in **Figure 7a-d**. These percent
differences are calculated using (1)

(1) Percent Difference$= \frac{(E_1 - E_2)}{\frac{1}{2}(E_1 + E_2)} \times 100$

where $E_2$ are the lidar observations and $E_1$ are the respective ozone values from the various platforms in
**Figure 6**.
The percent differences in **Figure 7a** indicate the GEOS-CF model from 20-45 km generally represents
the lidar observations, but are generally 0-10 % lower in abundance. The percent differences in **Figure**
**7b** indicate OMPS-LP is also representing the ozone maxima and altitude above 25 km. There are larger
differences below 20 km, which indicates the OMPS-LP retrieval is generally underestimating he ozone
abundance below 20 km as shown in the profiles in **Figure 5**. The percent differences in **Figure 7c**
indicate the MLS data, especially that within the 20-40km region, perform quite well as compared to the
lidar observations. The percent differences in **Figure 7d** indicate the TROPOMI retrieval is generally
over representing the ozone concentrations throughout the atmosphere, which worsens within the
troposphere.  In all cases, the most variability in the differences occur within the active region from 10-
20 km that is driven by the dynamical tropopause height and lower stratospheric ozone abundance.



## 5  Total Column Ozone

Similar to the troposphere, to better understand to what extent the vertical distribution of ozone impacts the atmospheric column, **Figure 8 (top panel)** shows the various platforms and their retrieved total column ozone. For this analysis, the GEOS-CF, lidar, OMPS-NM, TROPOMI (GODFIT) are shown, in addition to local ground-based measurements from a Pandora instrument and Brewer. The total column values range from 230-300 DU throughout the campaign period, with the median total column ozone of 271 DU. With the previous analyses from Sec 3.2, this indicates the median total tropospheric column of 33 DU and 0-2km boundary layer column of 6 DU result in percentages of the entire ozone column of 12% and 2.3%, respectively.  Similar to the full tropospheric ozone columns, larger total ozone columns were observed towards the end of the TROLIX-19 period, suggesting this variability was partly due to a larger abundance of ozone in the lower stratosphere.

**Figure 8 (bottom panel)** shows the various platforms as a percent differences from the model. In general, the various platforms are all within 10 % of each other, with most differences being within ±5%. This analysis emphasizes the stability and maturity of the Pandora and Brewer systems for monitoring the total column ozone amounts. Interestingly, the double maxima feature in vertical ozone distribution in the stratosphere (with local minima between) described in Sec 4.1 on 21 Sep does not severely impact the total column ozone.

## 6 Conclusions

This work has highlighted the various differences in retrieved ozone quantities during the TROLIX-19 campaign. This has emphasized the importance of ground-based ozone lidars and other measurements


in understanding the vertical variability of ozone and how it relates to the column reduction. This work
also shows the first effort to directly resolve both tropospheric columns and 0-2km ozone columns from
the NASA TROPOZ lidar. Other TOLNet lidars are able to perform this data reduction and future work
will be to expand this effort to the other TOLNet locations. This work indicates the level of
performance of the GEOS-CF modeling system as compared to the other platforms, which ultimately
performs extremely well both in the stratosphere and within the troposphere, as emphasized in **Figure 6**
and **Figure 7**. In looking towards the NASA TEMPO mission, this work indicates that the GEOS-CF is
an appropriate choice for the a priori profiles for the TEMPO ozone retrievals. This work shows the
TROPOMI ozone profile products are able to accurately reproduce ozone quantities in the lower
troposphere at various atmospheric levels. In particular, **Figure 3** shows promising results that indicate
the TROPOMI satellite observations compare well with the observations from ground-based
measurements (lidar, sonde) of specific elevated ozone features.
The CESAR Observatory continues to be a critical landmark for campaigns that revolve around
atmospheric composition measurements for satellite validation and evaluation beyond this effort, such
as CINDI and CINDI-2 (Kreher *et al.*, 2020; Tirpitz *et al.,* 2021). As the European Commission (EC) in
partnership with the European Space Agency (ESA) continues to launch tropospheric composition
satellites, including the upcoming geo-stationary Sentinel-4 satellite, we expect this observatory will
continue to host and maintain critical atmospheric sampling for future validation efforts.





Data Availability.
1.  MLS ozone profiles can be downloaded from the NASA Goddard Space Flight Center Earth

Sciences Data and Information Services Center (GES DISC; Schwartz et al., 2020,

https://doi.org/10.5067/Aura/MLS/DATA2516,  last access: 29 March 2022).

2.  The Pandora data is available at the Pandonia Global Network Archive ( http://data.pandonia-

global-network.org/Cabauw/, last access 29 March 29, 2022).

3.  The OMPS LP version 2.5 ozone profiles can be downloaded from the NASA Goddard Space

Flight Center Earth Sciences Data and Information Services Center (GES DISC;

at https://doi.org/10.5067/X1Q9VA07QDS7 (Deland, 2017, last access: 29 March 2022).

4.  The tropospheric ozone lidar data used in this publication were obtained from the Cabauw

Experimental Site for Atmospheric Research (CESAR) as part of a campaign involving the

Network for the Detection of Atmospheric Composition Change (NDACC) and NASA's

00        Tropospheric Ozone Lidar Network (TOLNet) and are publicly available (https://www-

01        air.larc.nasa.gov/cgi-bin/ArcView.1/TOLNet?NASA-GSFC=1, last access: 29 March 2022).

5.  The ozonesonde and Brewer data used in this publication were obtained from the De Bilt, NL

03        site as part of a campaign involving the Network for the Detection of Atmospheric Composition

04        Change (NDACC) and are publicly available (ftp://ftp.cpc.ncep.noaa.gov/ndacc/station/debilt/,

05        last access: 29 March 2022).

6.  The stratospheric ozone lidar data used in this publication were obtained from the Cabauw

07        Experimental Site for Atmospheric Research (CESAR) as part of a campaign involving the



Network for the Detection of Atmospheric Composition Change (NDACC) and are publicly
available (ftp://ftp.cpc.ncep.noaa.gov/ndacc/station/cabauw/, last access: 29 March 2022).
7. The TROPOMI TOPAS Ozone Profile data and source codes are available upon request from
Nora Mettig (mettig@iup.physik.uni-bremen.de) or Mark Weber (weber@uni-bremen.de). The
L1B version of the S5P data is available upon request to the S5P Validation Team.
8. The Tropospheric Ozone Column from OMPS-NM/MERRA-2 Daily measurements data are
available upon request from Jerry Ziemke (Jerald.r.ziemke@nasa.gov).
9. The NASA GEOS-CF simulations are available at the data sharing portal
(https://portal.nccs.nasa.gov/datashare/gmao/geos-cf/v1/forecast/, last access 29 March 2022).
Author contributions. JS drafted the original manuscript. JS, LT, GS, and TM deployed and operated
the NASA ozone lidars and provided expertise on use of measurements. NM, AR, and MW provided
TOPAS ozone profile data and guidance on how best to use the measurements. AA and KK provided
overall context as principal investigators of the TROLIX-19 campaign and coordinated science team
meetings to foster this collaboration.  KEK provided GEOS-CF data and insight on its use in this work.
MA, AP, MvR, and PV provided expertise and data for the ground-observations for ozonesondes,
Brewer, and historical data for the Cabauw site. JZ provided data for the OMPS-MERRA-2
troposphere column data. NK provided Aura MLS, OMPS-LP merged data and further insight into the
use of the data.
Competing interests. The authors declare that they have no conflict of interest.





Disclaimer. Publisher's note: Copernicus Publications remains neutral with regard to jurisdictional

claims in published maps and institutional affiliations.

Acknowledgements. NASA data has been provided through the Tropospheric Composition and Upper

Atmosphere Research Programs. We acknowledge all additional data providers and their funding

agencies for performing regular measurements and retrievals.

36

37
38

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

**Table 1: Instrument platforms, associated products, and short description used in this work during the TROLIX-19**
**campaign.**

| Instrument | Products | Platform | Description |
|---|---|---|---|
| GSFC TROPOZ [NASA] | Profiles [0.2 – 18 km] | Ground-based Lidar | 10 min integration; 30-90-min avg around ECC or Satellite Overpass |



| | | | |
|---|---|---|---|
| GSFC STROZ [NASA] | Profiles [15 - 48 km] | Ground-based Lidar | ~2-4-hr avg between (20-23 UT) |
| ECC Ozonesondes [KNMI] | Profiles [0 – 33 km] | Balloonborne | Balloonborne, Launched at 12 UT from De Bilt (~30 km from Cabauw) on 4 days |
| Pandora [NASA/KNMI] | Column [TCO] | Spectrometer | L2 Pandora 118s, Data Used has QC/QA Flags = 10 |
| Brewer [KNM] | Column [TCO] | Spectrophotometer | L2 Brewer #189m, MKIII, Located in De Bilt |
| S5P/TropOMI [ESA] | Column [TCL] | Satellite | L2 TOPAS Product, Overpass between 12-14 UT (5.5x3.5 km, nadir) |
| S5P/TropOMI [KNMI] | Column [TCO] | Satellite | L2 GODFIT v4 TO3 Product, Overpass between 12-14 UT (5.5x3.5 km, nadir) |
| OMPS [NASA] | Column [TCO] | Satellite | L3 NM Product, Version 2, Daily Overpass between 12-14 UT (50x50 km, nadir) |
| OMPS-LP [NASA] | Profiles [12-60km] | Satellite | Merged L2 v2.5 Daily Merged Product, Overpass between 12-14 UT (1km vertical bins) |
| OMPS/MERRA-2 [NASA] | Trop. Columns | Satellite/Assimilation | L4 Derived Product, OMPS-NM daily Overpass, MERRA-2 |
| AURA MLS [NASA] | Profiles [12-60km] | Satellite | Merged L2 v5 Daily Daytime/Nighttime Products, Overpass between 12-14 UT (1km vertical bins) and 01-03 UT. |
| GEOS-CF [NASA] | Profiles [0-80km] | Global 3-D CCMM | 1-Hr, 72 lev, Met. Replay, (25x25km) gmao.gsfc.nasa.gov/weather_prediction/GEOS-CF/ |


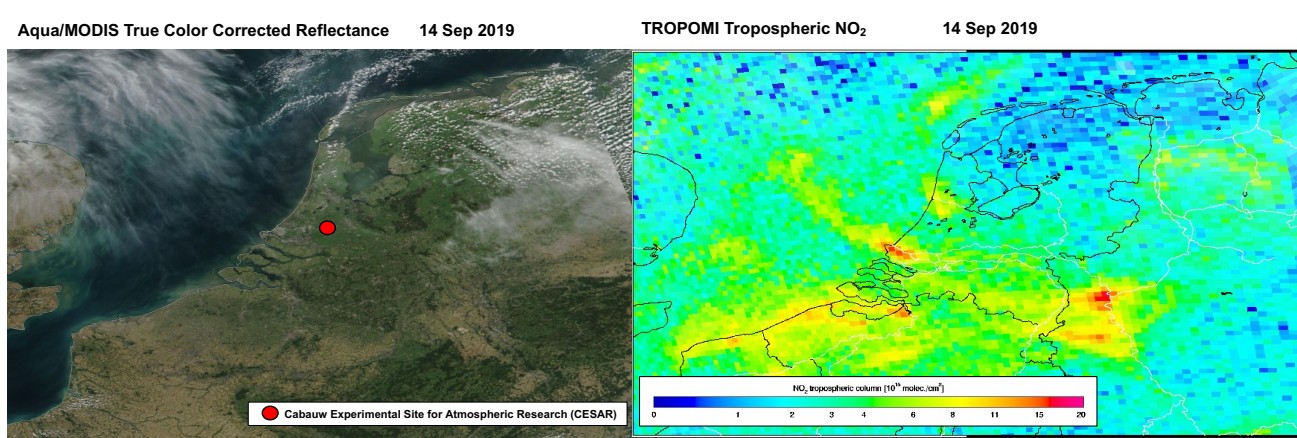


**Figure 1: Aqua/MODIS True Color Corrected Reflectance (left) and TROPOMI Tropospheric NO₂ (right) for 14 September 2019. The CESAR site is indicated in the image on the left.**



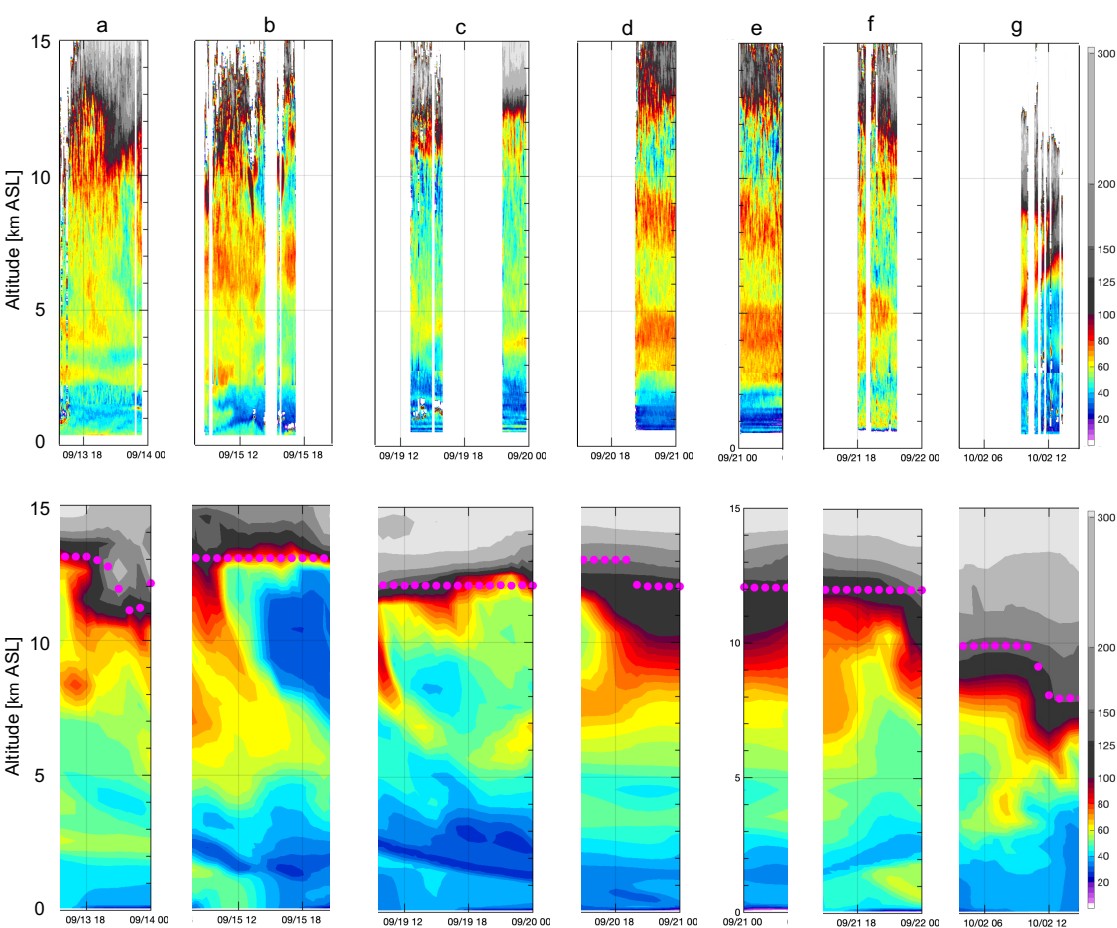


**Figure 2: Cloud screened TROPOZ lidar retrievals (top panel) and the corresponding GEOS-CF model output (bottom**

**panel) from the closest model grid cell to the CESAR observatory during TROLIX-19 for a) 13 Sep 14-00 UTC, b) 15**

**Sep 09-21 UTC, c) 19 Sep 10-00 UT, d) 20 Sep 16-00 UT, e) 21 Sep 0-3 UT, f) 21 Sep 16-00UT, and g) 02 Oct 04-14 UT.**

**Pink dots are overlaid to indicate the simulated tropopause altitude based on a blended estimate (TROPPB).**

55
56
57



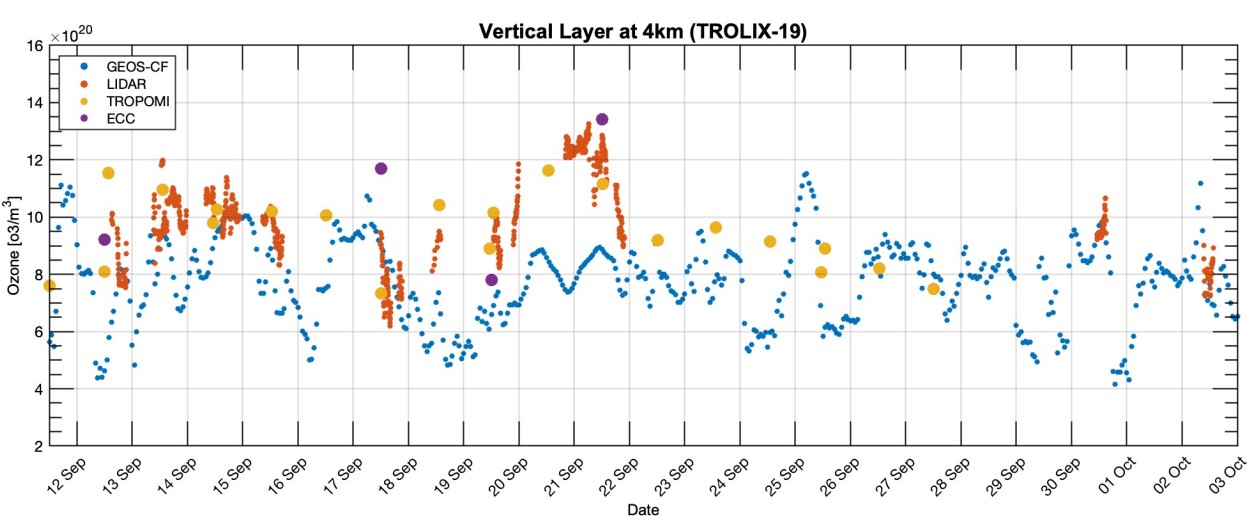

**Figure 3: Ozone number density values for the TROPOZ lidar, GEOS-CF mode, TROPOMI and electro-chemical cell (ECC) ozonesondes at the 4km layers/levels.**





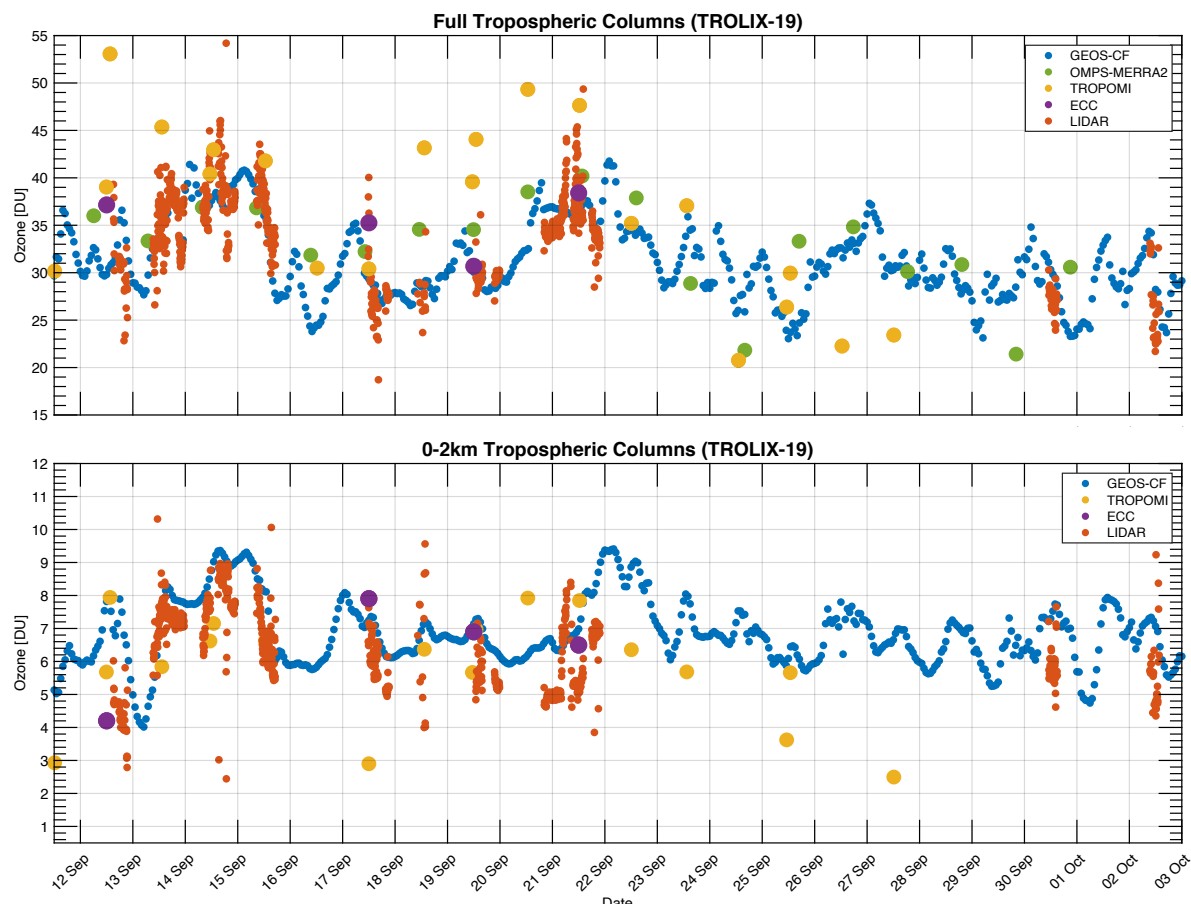


**Figure 4: Full troposheric columns (top panel) and 0-2km tropospheric columns (bottom panel) calculated from GEOS-CF, OMPS-MERRA2 (full column only), TROPOMI, Lidar and ECC.**







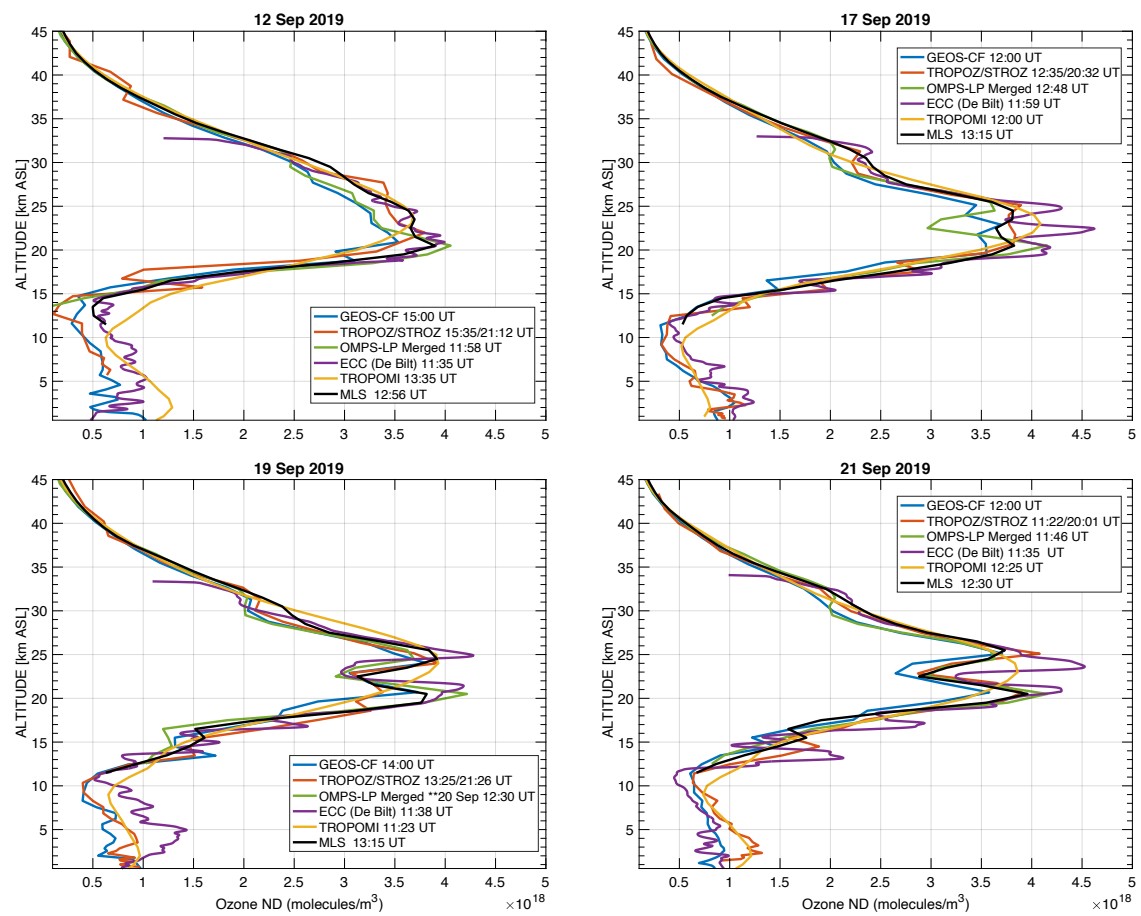


**Figure 5: GEOS-CF, Lidar, OMPS-LP, ECC, TROPOMI, and MLS ozone profile comparisons for 12 Sep, 17  Sep, 19 Sep, and 21 Sep 2019. These days were selected as days within the campaign that had an ECC launch from De Bilt.**








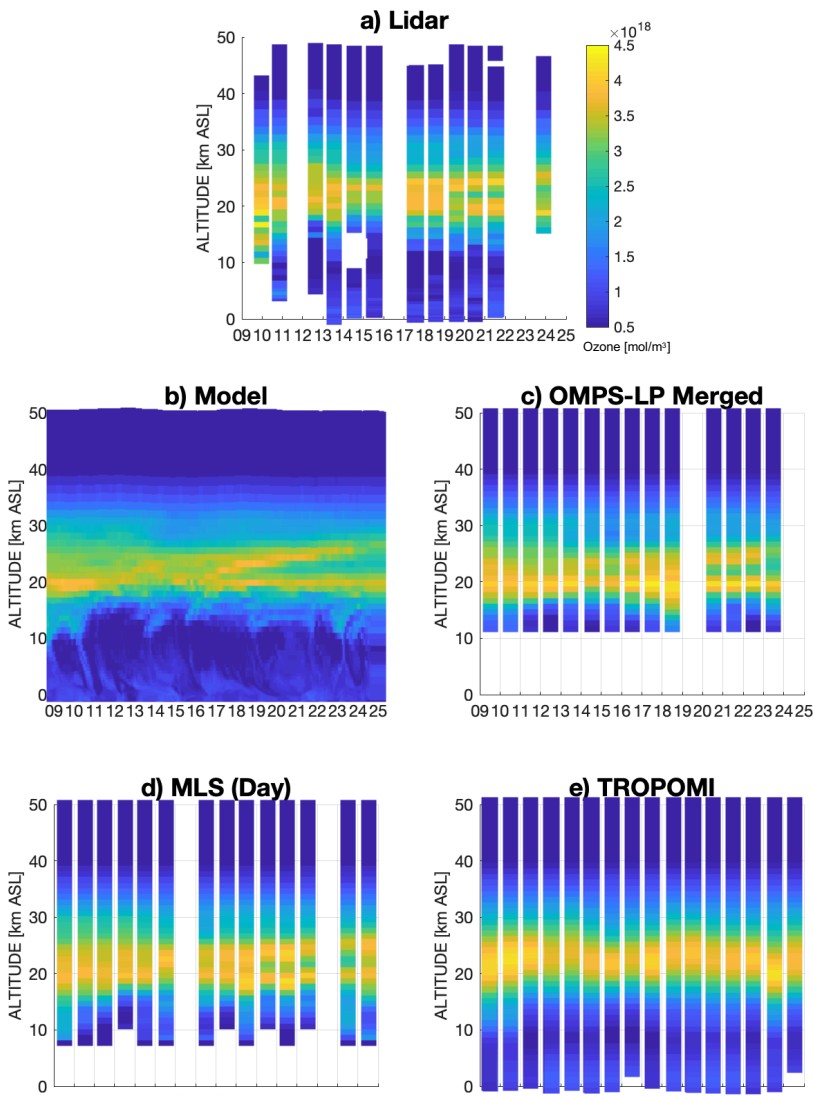

**Figure 6: Ozone number densities across all platforms for the TROLIX-19 time period from the hybrid lidar dataset**
**(Figure 6a), GEOS-CF (Figure 6b), OMPS-LP (Figure 6c), MLS (Figure 6d), TROPOMI (Figure 6e).** The x-axis as
day of September 2019.


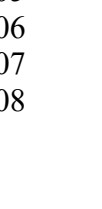


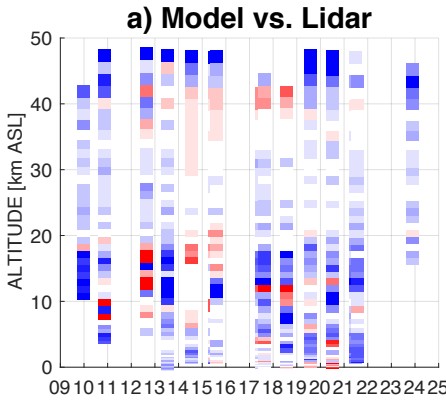
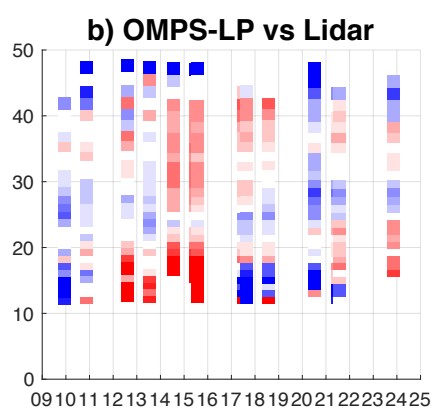
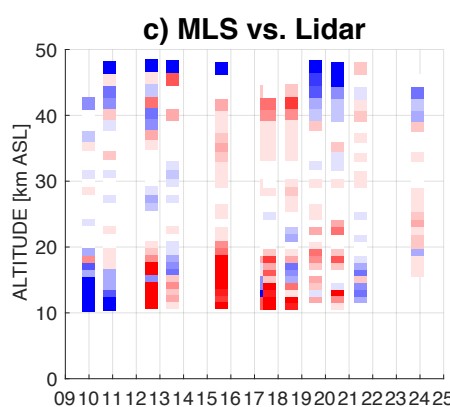
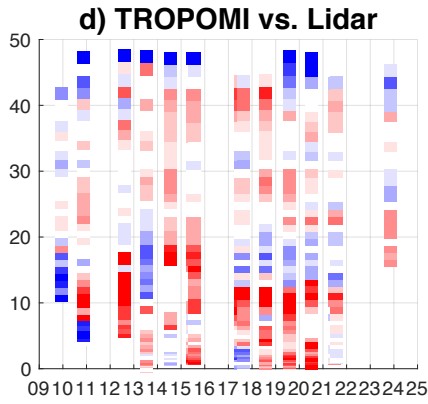


**Figure 7: Differences in ozone number densities across all platforms for the TROLIX-19 time period for Model (Figure**

**7a), OMPS-LP (Figure 7b), MLS (Figure 7c), and TROPOMI (Figure 7d). The x-axis as day of September 2019.**



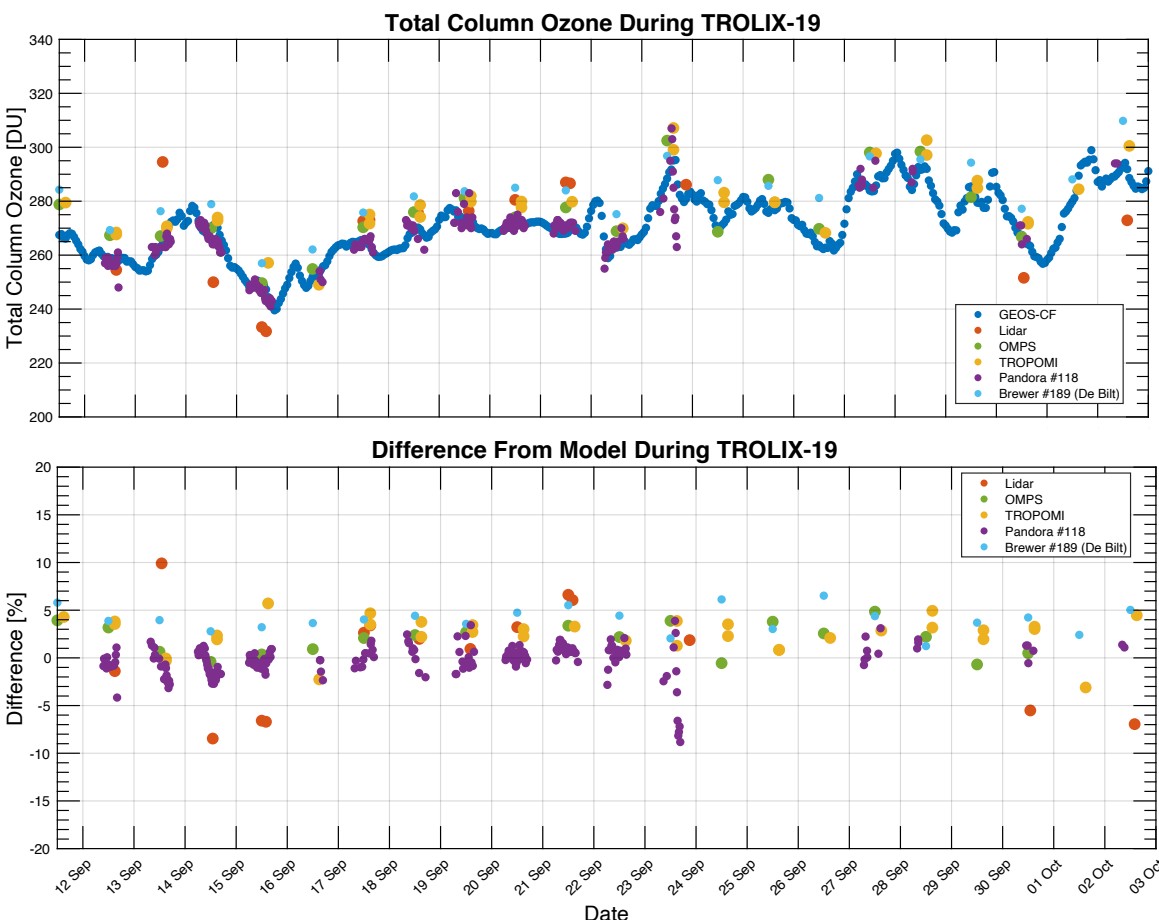

19  **Figure 8: Total Ozone columns (top panel) and percent differences (bottom panel) as compared to the model**

20  **observations for GEOS-CF, lidar, OMPS, TROPOMI, Pandora, and Brewer.**

