# Peer review of "Tropospheric and Stratospheric Ozone Profiles during the 2019 2 TROpomi vaLIdation eXperiment (TROLIX-19)"

_Atmospheric Chemistry and Physics, 2022_

## Author Comment (AC1)

The paper submitted to ACP by J.F. Sullivan and co-workers is a measurement report of data collected during a joint NASA-KNMI field experiment designed to validate ozone retrieval by the TROPOMI spaceborne mission. On 11 days, lidar O3 observations in the troposphere and the stratosphere are carried out for several hours at the Cabauw site. The two NASA lidars are indeed very valuable tools for the assessment of TROPOMI. Observations of total ozone column (TOC), stratospheric limb sounders (MLS), and nearby ECC launched at De Bilt are also analyzed in this work, in addition to the daily ozone simulations of the GEOS-CF model. The time evolution of TOC and tropospheric O3 partial columns (0-2km, 0km-Tropopause level) is used to assess TROPOMI observations and GEOS-CF model ozone mapping. Considering the value of satellite validation exercises and comparisons between model and observations, the paper is appropriate for publication in ACP (or AMT considering the focus on experimental data analysis). It is well written with some remaining errors in figure legends. I fully agree that there is a general good agreement between the different observations and that the GEOS-CF model performs quite well.

AR: Thank you for understanding the value of utilizing many different platforms to better evaluate the satellite products and their physical interpretation. We hope to continue using the lidars, models, and other ground-based observations to evaluate future satellites as well. We also agree this is well within the scope of ACP (especially the special issue) and think has broad enough scope to be quite impactful.

My main criticism would be that detailed discussions of the differences observed in the troposphere are sometimes missing and that these differences need to be acknowledged in the conclusions: TROPOMI overestimate in the lowermost stratosphere (10-15 km), GEOS-CF low performance during the 9/20-21 episode (not only at 4 km but also below the tropopause leading to small differences of tropospheric columns without a good agreement in the vertical profile), ECC/lidar positive differences in the free troposphere on 9/12,17,21 (De Bilt/Cabauw spatial differences ?).

AR: We appreciate the thoughtful review and have answered the below comments that should be more than adequate to alleviate any concerns.

Specific comments

Line 45 : Lidar is also sensitive to cloud cover. Are missing lidar data during the campaign related to cloud cover ?

AR: Yes and this has now been mentioned in the paragraph beginning on Line 116.

Fig. 1 : Fig. 1 is not referenced in the text. What is the purpose of this figure ? Please also show De Bilt and Cabauw on the NO2 map.

AR: Figure 1 is now reference in line 50. The purpose of this figure is to provide the read with geographic context for the site. Rather than focus on a specific day in the original submission (Sep 14, 2019), the image now serves as a monthly mean for September 2019 (the TROLIX-19 period).

[Figure]

Line 108 : Addition of lidar O3 measurement accuracy and vertical resolution would be useful in Table 1. Since partial tropospheric columns and TOCs are discussed in section 3.2 and 5, the expected accuracy of the lidar retrieval on these columns would help.

AR: The below text has been added to the paper near line 125 to better characterize the uncertainty estimates.

A brief description and community standardized definitions of the uncertainty budget of the lidar measurements presented in this paper can be found in Sullivan *et al.,* 2014, Leblanc *et al.,* 2016 and Leblanc *et al.,* 2018. The maximum statistical uncertainties for the two GSFC lidars vary from night to night depending on atmospheric conditions and laser power fluctuations. They are mostly within 10-20% for 5 min and 5-8% for 30 min integrations throughout the atmosphere within their measurable ranges although they are different at the same altitude due to laser performance and telescope/detector efficiency differences.

Line 151 : Give a more recent reference i.e. Smit and Thompson 2021 GAW report 268 describing ASOPOS 2.0 error calculations.

AR: This reference has been added and we appreciate the attention to detail relating to contemporary references on this subject matter.

Line 171 : What is the expected difference when using the 2.5 PVu instead the 3 PVu tropopause definition taken for the lidar tropospheric ozone column calculation ?

AR: The tropopause height was calculated from the GEOS-CF and was then used consistently for all measurements in the trop. ozone column calculations. We have confidence that this is appropriate by visually comparing the tropopause heights compared to the model and lidar in Figure 2 and later on with ozonesonde values. Within the community, there are several definitions of the tropopause height, and we have done our best effort to stay consistent with the 2.5 PVU value and have described the method used to calculate this in the literature.

Line 206 : What is the typical GEOS-CF model vertical resolution in the UTLS ?

AR: There are 72 total vertical layers in the GEOS-CF model, scaled by atmospheric pressure From 10 mb to 375mb there are 20 vertical layers. In general this is roughly ~1.1-1.2 km and a visual reference is shown in Fig. 3 here :https://acp.copernicus.org/articles/17/1417/2017/acp-17-1417-2017.pdf

You can find much of the details (also cited in paper as Keller et al., 2021) pertaining to the GEOS-CF system here:
https://agupubs.onlinelibrary.wiley.com/doi/10.1029/2020MS002413

Line 225 : Add 2f in the list  of panels with large lidar/model differences.

AR: Done.

Line 229 : It is also true on 9/13-14.

AR: Added.

Line 233 : What is the thickness of the 4-km layer used for plotting ozone in Fig. 3 ?

AR: The layer was calculated to match the closest vertical layer of the GEOS-CF for consistent intercomparison. The top of model layer 53 is at 3.94 km (~605 mb, for the specific files used in this work) and the top of model layer 54 is 4.44 km (~570 mb, for the specific files used in this work). This is roughly a 500m or 40mb layer thickness.

Figure 2 : Ozone unit is not specified in the caption or color scale. Please add the pink tropopause altitude on the ozone plot and the vertical limits of the 4-km layer used in Fig. 3.

What is the time end of panel e ? This plot is very nice. It's a shame not including the two days with ECC at De Bilt on 9/12 and 9/17.

AR: The ozone unit in Figure 2 has been added – Ozone mixing ratio [ppbv]. Also the tropopause heights have been added to the ozone lidar measurements. The details of the layer thickness have also been added to the caption.

Line 243 : The O3 vertical structure in the UTLS is also missed by the model (no low O3 at 11 km). It is then likely that the mesoscale ozone 3D transport in the frontal system is not very well resolved by the model for this specific event.

AR: A comment reflecting this has been added to the revised manuscript starting near Line 286.

Line 246 : Why are 9/17 ECC ozone concentrations significantly larger than model, TROPOMI and lidar O3 values ? Did you see any horizontal O3 gradients near Cabauw in the model or TROPOMI mapping ?

AR: Below is a quick plot of the 9/17 ECC (however, not added to current manuscript). You can see the sampled ozone, RH, and wind speed in their respective units. There is no sign of any pump current overheating or other QC/QA issues that may cause invalid data within the sonde chamber. In particular, this feature at ~4km near 1.2e21 mol/m3 is associated with a much drier air mass and is notable traveling upwards of 30 m/s during the sampling period. The wind direction was consistently coming from the WNW during this period.
At this speed and direction, the sonde (already launched at De Bilt) may have just been sampling a similar but not identical air mass. The differences begin to decrease in this profile compared to the lidar near about 11km (as shown in the current manuscript Figures).

[Figure]

Line 271-274 : This discussion is unclear. The difference between the tropospheric columns are in fact not too bad but are fortuitous because the effect of the missing layer at 4 km is cancelled by the high model ozone at 11 km. Please reformulate this part of the discussion.

AR: A clarification regarding this has been added to the text near Line 408.

Line 279 Figure 4 low panel (not top panel)

AR: Done.

Line 282 I do not understand this sentence.

AR: A clarification regarding this has been added to the text and discussion in the same paragraph has been added from the previous comment (starting near line 408).

Line 285 What are the reasons for low 0-2 km column in some lidar data on 9/12 and 14 and in TROPOMI data on 9/17 ?

AR: These values were driven by cloud contamination in the lower levels of the troposphere for all observations.

For the 09/12 and 09/14 lidar data, this is due to some data points being cloud contaminated. A stricter (and more manual) cloud screen using lidar backscatter SNR has been implemented now.

For the low TROPOMI data on 09/17 (and by extension a few other days in the second half of the domain) are also likely due to cloud interference. There were several TROPOMI pixels that were no retrieved because they did not match the quality requirements (noise too large). By using the OMPS reflectivity data, I have screened additionally for the tropospheric and 0-2km column data in Figure 4. This was changed from a 0.8 reflectivity to 0.6 reflectivity and a note has been made in the caption.

Line 288 A discussion on the TROPOMI retrieval of the partial column is needed in this section. While the 0-2 km columns remain within the range of the diurnal variability, several full tropospheric columns (9 /12,13, 18,19,20) are well above this diurnal variability. It is likely related to the limited vertical resolution in the UTLS. The TROPOMI overestimates are also quite clear in the ozone profiles shown in Fig. 5.

AR: This is an important point, thank you for this. A short discussion has been added, and the reader is directed to the Mettig et al., work that systematically goes through the retrieval from first principles within the spectral range of the ozone profile retrieval (https://amt.copernicus.org/articles/14/6057/2021/amt-14-6057-2021.pdf).

Line 299 Reading this sentence, I am not sure which altitude range is critically needed for monitoring of the column

AR: This has been specified in the context of the previous comment.

Line 308 What is OMPS-LP ? The stratospheric MERRA-2 profile used for the OMPS tropospheric ozone column calculation ? Do hybrid lidar profiles combine daytime TROPOZ data with nighttime STROZ-LITE data or are nighttime profiles only considered in the hybrid version ? Please clarify this point.

AR: From Section 2.3.1, this is the OMPS Limb Profiler ozone profile product and a clarifying statement has been added to start Sec 2.3.1.

The vertical distribution of ozone in the stratosphere and lower mesosphere is obtained from the OMPS Limb-Profiler (LP) sensor on the Suomi-NPP satellite merging the UV (29.5-52.5 km) and VIS (12.5-35.5 km) bands to provide a full profile from 12.5km to 52.5km (Kramarova *et al.*, 2018).

AR: I have specified that this is daytime or nighttime TROPOZ data, and only nighttime measurements for the STROZ. This was ultimately manually selected each day since this was such a focused intensive campaign and we were able to do so. Laser performance and cloud considerations also drove the decisions.

Fig. 5 I believe that the titles of the bottom panels are wrong otherwise Fig.3 and Fig.5 are not consistent.

AR: Great catch – the ozonesonde read file was pointing to opposite directories for those two days (so the titles were correct, but the ECC profile is swapped). This has been corrected and actually shows a much better comparison to the lidar and model, while simultaneously highlighting the feature described in Fig 3.

Line 320 Comparisons of the stratospheric profiles are extensively discussed while the general agreement is quite good in the stratosphere. The discussion of the tropospheric differences is however limited while  there are some interesting differences (TROPOMI in the UTLS, lidar ECC differences).

AR: A clarification regarding this has been added to the text throughout Lines 352-373 and is now reiterated in the concluding remarks.

Fig. 7 Color scale is missing

AR: This has been added and this was a major oversight to miss on initial submission.

Line 339: Why do you say that OMPS-LP underestimates O3 concentrations at altitudes below 20 km ? I see both positive and negative differences. Regarding TROPOMI, it looks like TROPOMI versus MLS is better between 10 and 20 km than the large lidar/TROPOMI differences in the UTLS. Is it related to similar UTLS vertical resolutions for MLS and TROPOMI ?

AR: A clarification regarding this has been added to the text throughout Lines 352-373 and is now reiterated in the concluding remarks.

Line 358: A 7% difference on TOC is already quite significant. What are the reasons for the low TOCs given by the hybrid lidar retrieval on 9/14-15 or by Pandora on 9/23 ?

In looking further at the mismatch in lidar total column ozone values, we discovered the calculation was performed and plotted with the corresponding TROPOZ integration time of day which was predominantly during the daytime observations. For this specific section describing total column ozone, the majority of that information is coming from the STROZ LITE nighttime data. Therefore, more optimal merges (particularly in the UTLS) were performed to include as much STROZ data as possible and shift the red data points to a more representative time of the lidar observations. This results in a much closer comparison to the observations as compared to the previous figure for the time in question.

Pandora flags equal to 10 (purple filled dots) and 11 (purple open dots) are shown. On 09/23 there were many observations mixed with both flags throughout the day, likely due to cloud contamination. Only data with the highest QC/QA level (i.e. 10) is plotted in the updated final figure. This results in less data overall.

[Figure]

Line 371: GEOS-CF indeed performs well reproducing the ozone downward transport in the UTLS, but a sentence could be added about GEOS-CF failure to resolve some high resolution laminae related to specific mesoscale ozone 3D transport from the UTLS, e.g. 09/19-20.

AR: A clarification regarding this has been added to the text starting near Line 288.

Line 374: The overestimate of the TROPOMI retrieval between 10 and 15 km needs to be mentioned in the conclusions.

AR: A clarification regarding this has been added throughout the previous reccomendations and line numbers above as well as additional context in the conclusions.

We thank you for such a thoughtful review (including a few typos/errors) and know the manuscript has improved greatly from your suggestions.

Anonymous Referee #2

Review of "Measurement Report: 1 Tropospheric and Stratospheric Ozone Profiles during the 2019 TROpomi vaLIdation eXperiment (TROLIX-19)" by J.T. Sullivan et al.

In this manuscript the authors describe the many ozone measurements made from multiple platforms during the TROpomi vaLIdation eXperiment (TROLIX-19) campaign in fall 2019. The campaign was designed to support satellite validation efforts with the emphasis on understanding the vertical profile retrievals of ozone. Instrumentation included ozone lidars, pandora spectrometer, Brewer Spectrophotometer and ozonesondes. Satellite data used include OMPS, MERA-2, MLS and TROPOMI. In addition, measurement data is compared with GEOS-CF model output.

The analysis focuses on two main goals; one to evaluate ozone retrievals in relation to current and future (TEMPO) satellites. And the combination of the tropospheric ozone lidar and the stratospheric lidar providing hybrid ozone profiles from ~0.2 km to ~50 km.

The article is clearly written and provides a comprehensive presentation of data from a number of measurement platforms. With figures and discussion of the comparisons across measurement platforms and model within the full column ozone and 0-2 km tropospheric column ozone. The study makes several important observations as to the structure of the ozone within the atmospheric column and the differences in instrument/model performances.

AR: Thank you for understanding the value of utilizing many different platforms to better evaluate the satellite products and their physical interpretation. We hope to continue using the lidars, models, and other ground-based observations to evaluate future satellites as well. We also agree this is well within the scope of ACP (especially the special issue) and think has broad enough scope to be quite impactful.

Specific comments:

I would suggest that the conclusions section needs to be expanded upon. This section emphasizes the importance of observations and for the site itself (which I agree on) but I would like to see more quantitative evaluation of the data here to back up the statements, adding some percentage differences etc would make this an easy reference source for future readers. For example, there is a statement (line 74-75): " TROPOMI ozone profile products are able to accurately reproduce ozone quantities in the lower troposphere…" with reference to Figure 3 but this only shows TROPOMI compared with observations vertical layer at 4km. In addition, Figure 7 indicates that TROPOMI generally overestimates, especially within the troposphere. Expanding the conclusions to include some of the quantitative results would help to firm up the concluding statements.

AR: The conclusion section has been broadened beyond the importance of observational platforms. We have added several discussion clarification on TROPOMI overestimations per the previous reviewer that are intended to improve the communication surrounding this issue.

Figure 7 was intended to be the quantitative resultant figure that could be referenced or looked up easily and this has been elaborated on in the conclusions. We feel this figure better illustrates both the temporal (over the course of the campaign) and vertical differences observed in the retrievals and reducing that down to a simple number or XX % may underserve the push for the vertical profiling needed for these types of efforts. (And hopefully the colorbar now added helps the reader too!)

Figure 1. Add the CESAR site also be added to the image on the right or add lat/lon information to maps.

AR: CESAR has been added.

Figure 7. Needs a color bar with values for the differences in ozone number densities

AR: Done, thanks!

Reviewer 3:

Review of "Measurement Report: Tropospheric and Stratospheric Ozone Profiles during the 2019 TROpomi vaLIdation eXperiment (TROLIX-19)" by J.T. Sullivan et al, submitted to ACP

In this manuscript, a measurement report is given on the various ground-based measurements deployed during the TROLIX-19 field campaign in Cabauw, Netherlands during September 2019. The focus of the campaign and this paper was to provide comprehensive validation of the TROPOMI ozone profile retrievals using ozone lidar measurements, the measurement focused on in this paper, as well as ozonesondes, Pandora, Brewer, etc. Ozone lidar measurements were also compared against other satellite and model datasets, such as OMPS (satellite), MLS (satellite), and GEOS-CF (model). The authors also analyzed the temporal variability of full tropospheric and 0-2 km ozone columns using GEOS-CF, TROPOMI, ozonesondes, and ozone lidar.

The article provides a comprehensive analysis of the measurements and ancillary datasets used to demonstrate the variability of tropospheric and stratospheric ozone profiles during TROLIX-19. The authors clearly and concisely state the impact of the ground-based measurements have on understanding the capabilities and limitations of current polar orbiting and future geostationary ozone retrievals.

Specific comments:

Line 95: It would be worth mentioning how many stations are in the network

AR: Done.

Line 145: Correct spelling of campaign

AR: Done.

Line 147: Specify which radiosonde manufacturer was used and the specific model

AR: Done.

Line 159: Put space between "are" and "used"

AR: Done.

Figure 2: Add label to the color bar, include units

AR: Done.

Line 233: It can be assumed that 4 km is chosen to investigate the model and observation difference observed between 3-5 km on 20-21 September. However, it should be specifically mentioned at the beginning of this paragraph why 4 km is chosen.

AR: A clarification has been added.

Line 279: Correct to say Figure 4, bottom panel

AR: Done.

[revised manuscript text omitted]